# The impact of angles of insonation on left and right ventricular global longitudinal strain estimation in fetal speckle tracking echocardiography

Thomas J. Nichting[1,2,3]☯*, Chantelle M. de Vet[1,2,3]☯, Myrthe van der Ven[1,3,4], Daisy A. A. van der Woude[1,2,3], Marta Regis[5], Ruud J. G. van Sloun[2,3], S. Guid Oei[1,2,3], Judith O. E. H. van Laar[1,2,3], Noortje H. M. van Oostrum[6]

1 Department of Gynaecology and Obstetrics, Máxima MC, Veldhoven, The Netherlands, 2 Department of Electrical Engineering, Eindhoven University of Technology, Eindhoven, The Netherlands, 3 Eindhoven MedTech Innovation Centre, Eindhoven, The Netherlands, 4 Department of Biomedical Engineering, Eindhoven University of Technology, Eindhoven, The Netherlands, 5 Department of Mathematics and Computer Science, Eindhoven University of Technology, Eindhoven, The Netherlands, 6 Department of Gynaecology and Obstetrics, University Hospital Gent, Gent, Belgium

☯ These authors contributed equally to this work.
* Thomas.Nichting@mmc.nl

## Abstract

### Objectives

Two-dimensional speckle tracking echocardiography has been considered an angle-independent modality. However, current literature is limited and inconclusive on the actual impact of angle of insonation on strain values. Therefore, the primary objective of this study was to assess the impact of angles of insonation on the estimation of fetal left ventricular and right ventricular global longitudinal strain. Secondarily, the impact of different definitions for angles of insonation was investigated in a sensitivity analysis.

### Methods

This is a retrospective analysis of a prospective longitudinal cohort study with 124 healthy subjects. The analyses were based on the four-chamber view ultrasound clips taken between 18+0 and 21+6 weeks of gestation. Angles of insonation were categorized into three groups: up/down, oblique and perpendicular. The mean fetal left and right ventricular and global longitudinal strain values corresponding to these three groups were compared by an ANOVA test corrected for heteroscedasticity.

### Results

Fetal left and right ventricular global longitudinal strain values were not statistically different between the three angles of insonation (p-value >0.062 and >0.149, respectively). When applying another definition for angles of insonation in the sensitivity analysis, the mean left

**Data Availability Statement:** The data that support the findings of this study are not publicly available due to restrictions e.g. their containing information

that could compromise the privacy of research participants. Data requests can be sent to the corresponding author or the ethics committee: METC Máxima Medisch Centrum. Contact details: Mw YIC de Haan en mw dr AM Nieuwesteeg, MSc Postbus 7777 5500 MB Veldhoven (The Netherlands) 040 888 9528 metc@mmc.nl.

**Funding:** The author(s) received no specific funding for this work.

**Competing interests:** The authors have declared that no competing interests exist.

**Abbreviations:** AoI, Angle(s) of Insonation; Fetal heart clip, Four-chamber view ultrasound clip; FR, Frame rates; GA, Gestational Age; GLS, Global Longitudinal Strain; IQR, Interquartile Range; LV, Left Ventricular; ROI, Region of Interest; RV, Right Ventricular; SD, Standard Deviation; 2D-STE, Two-dimensional Speckle Tracking Echocardiography.

ventricular global longitudinal strain value was significantly decreased for the oblique compared to the up/down angle of insonation (p-value 0.041).

## Conclusions

There is no evidence of a difference in fetal left and right ventricular global longitudinal strain between the different angles of insonation in fetal two-dimensional speckle tracking echocardiography.

## Introduction

Functional assessment of the fetal heart is proven to be helpful in the detection of common pregnancy complications such as fetal growth restriction and pregnancy diabetes [1–3]. Cardiac function can be assessed by two-dimensional echocardiography techniques [4]. However, as inherent to the use of Doppler velocity measurements, these techniques depend on the angle of insonation (AoI) [5]. Doppler-based measurements are difficult to obtain if the fetus is in an unfavorable position. Two-dimensional speckle tracking echocardiography (2D-STE) assesses cardiac function using myocardial deformation imaging (strain imaging) and has been considered angle-independent [6–9].

In 2D-STE displacement of speckles, scatters of the ultrasound beam by the tissue, are being tracked. These speckles yield a local acoustic fingerprint that can be tracked frame to frame during a cardiac cycle [10]. The speckles' position change compared to its original end-diastolic position in the myocardium, not in relation to the ultrasound beam [11,12]. Therefore, 2D-STE is considered to be angle-independent.

A recent study questioned the angle-independency of 2D-STE and demonstrated a significant difference between AoI of the fetal left ventricular (LV) global longitudinal strain (GLS) [13]. However, gestational age (GA) and the quality of the fetal four-chamber heart clip are known factors affecting GLS measurements, which could have influenced the study results [14–16]. Fetuses with various GA were included. Moreover, adequate quality four-chamber view ultrasound clips (fetal heart clip) were achieved in only 62% of cases for all AoI [13,14].

Assessment of the impact of AoI on GLS measurements is essential for reliable future research on this topic. To assess whether 2D-STE is angle-independent, adequate quality fetal heart clips, high FR and correction for GA is needed [8]. Therefore, the current study aimed to study the impact of AoI on fetal LV-GLS and right ventricular (RV) GLS values considering all these influencing factors.

## Material and methods

### Population

This is a retrospective analysis of a prospective longitudinal cohort study (NL64999.015.18). This study was performed in a tertiary care teaching hospital in the Netherlands from May 2018 until April 2019 [17].

The Board of the Medical Ethics Committee of Máxima MC, Veldhoven, The Netherlands, confirmed that the Medical Research Involving Human Subjects Act does not apply to the current retrospective study and granted a waiver for ethical approval (N21.054).

Healthy women from 18 years onwards with an uncomplicated singleton pregnancy between 18+0 and 21+6 GA and a normal second-trimester anomaly scan were included in

the prospective longitudinal cohort study. Exclusion criteria were women suffering from a systemic disease, including pre-existent diabetes mellitus, hypertensive disorders and an estimated fetal weight below the 10th percentile corrected for GA. GA was based on first-trimester crown-rump length measurement. Women who developed gestational diabetes or hypertensive pregnancy disorders, and those who gave birth to a neonate with a birth weight below the 10th percentile or with congenital or genetic abnormalities were excluded from the analysis as these factors may influence myocardial deformation values [2,3,10,18–21]. Confirmation of the exclusion criteria was completed 10 weeks after childbirth. Subjects not meeting the in- and exclusion criteria were removed from the cohort.

## Data acquisition and 2D-STE analysis

At least three fetal heart clips per fetus with a duration of three seconds have been recorded by an experienced obstetric sonographer (NvO, CdV) following a strict protocol [17]. A Philips Epiq W7 ultrasound system (Royal Philips N.V., Amsterdam, The Netherlands) with a 9-MHz linear transducer was used. 2D-STE analysis was performed offline on the raw uncompressed data of the fetal heart clips with TomTec Cardiac Performance 1.2 software (TomTec Imaging Systems, Munich, Germany) by a gynaecologist (NvO) with extensive experience in the assessment and performance of fetal heart ultrasounds and 2D-STE analysis. The experienced gynaecologist (NvO) selected the best one out of the three available fetal heart clips per fetus, based on fetal heart clip quality and FR, explained in detail below. 2D-STE analysis was performed on this fetal heart clip.

Fetal heart clips were considered adequate for 2D-STE when they consisted of a complete four-chamber view with sharp boundaries between the endocardium, lumen and the AV-valves, without the outflow tracts being visible or the presence of acoustic shadows or fetal movements. To ensure the attainment of adequate quality fetal heart clips, it is important to appropriately set the region of interest (ROI) and to use [8] optimal settings for image depth, width, and zoom box contributing to achieve maximum frame rates (FR) while minimizing potential speckle anisotropy impact on fetal 2D-STE [14]. Fetal heart clips with a FR lower than 80 Hz were not considered feasible and were excluded from the analysis [8]. The ROI also indicates the specific cardiac layer used for offline delineation. Within a 2D-STE analysis, transitioning the delineation from the endocardial to the epicardial border significantly affects global longitudinal strain (GLS) values [22]. Prior to offline delineation, the ROI was set, with preference given to the narrowest endocardial border as the default option [22].

## Angles of insonation

The AoI was defined as an end-systolic angle between the middle of the intraventricular septum and the ultrasound beam. Prospectively the AoI were divided into three categories: up/down, perpendicular and oblique. The up/down AoI was defined as an angle between $0 \pm 22°$ or $180 \pm 22°$, the perpendicular AoI was defined as an angle between $90 \pm 22°$ or $270 \pm 22°$, and the oblique AoI was defined as an angle between $45 \pm 22°$, $135 \pm 22°$, $225 \pm 22°$ or $315 \pm 22°$. Each position of the fetal intraventricular septum could be categorized in one of these three AoI.

A sensitivity analysis was conducted to investigate the impact of the AoI definition. In this analysis, we employed the definition proposed by Semmler et al. [13]. For the sensitivity analysis, we categorized the up/down AoI as an angle between $0 \pm 15°$ or $180 \pm 15°$, the perpendicular AoI as an angle between $90 \pm 15°$ or $270 \pm 15°$, and the oblique AoI as an angle between $45 \pm 15°$, $135 \pm 15°$, $225 \pm 15°$, or $315 \pm 15°$. Fetal intraventricular septum positions falling outside these angles were considered part of the transition zones and were excluded from the 2D-STE analysis. Fig 1 illustrates the different AoI categories when utilizing the various definitions.

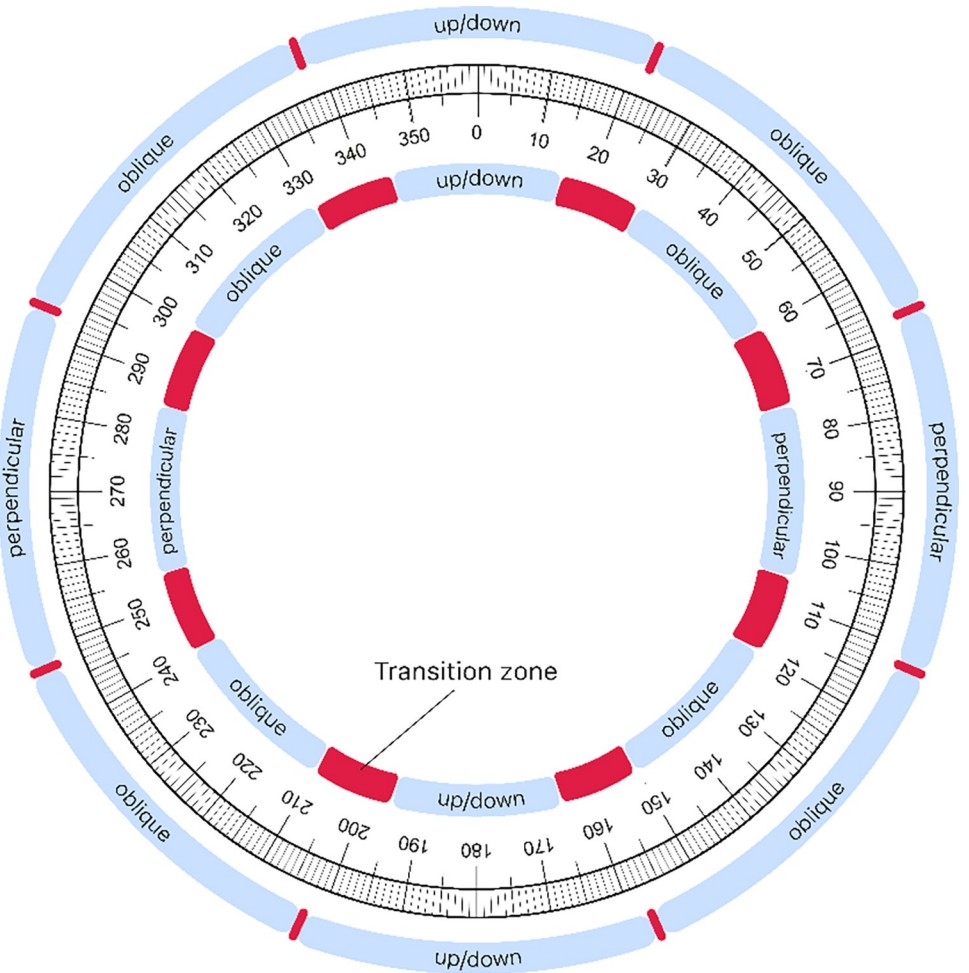

**Fig 1. Definitions for angles of insonation.**

The outer circle shows the definition for the different angles of insonation used for this study. The inner circle visualizes the definition for angles of insonation as used by Semmler et al. [13]. The red areas show the transition zones associated with both definitions.

## Statistical analysis

The normality of the data was assessed statistically per category using the Shapiro-Wilk test and qualitatively by inspecting histograms and normality plots. Descriptive statistics for continuous variables are presented as means and standard deviations (±SD) or medians and interquartile ranges (IQR) depending on the normality of the distribution. Categorical variables are presented as absolute numbers and percentages. To compare GLS values measured along the three different AoI, we fitted a one-way heteroskedastic ANOVA model [23]. This allowed us to account for differences in variability between the three AoI categories. The estimated marginal mean differences were also derived for each pairwise comparison. To explore the impact of the AoI definition, we performed a sensitivity analysis repeating the same analyses when the AoI were defined accordingly to previously published literature, including a transition zone between the AoI [13]. P-values below 0.05 were considered statistically significant. Statistical analyses were conducted with a statistician (MR) using SAS software, Version 9.4 of the SAS System for Windows.

## Results

Fig 2 illustrates the flowchart depicting the selection and categorization process for the included fetal heart clips. Out of the initially available 372 fetal heart clips (three per fetus), 124 clips (33.3%) were selected. Following a quality assessment, 8 clips (7%) were excluded due to low quality, resulting in 116 clips (93%) with adequate quality. The median gestational age of the 116 fetuses included in the analysis was 19+2 weeks (range: 18+0–21+6 weeks). Among these clips, 69 (59.5%) had an oblique AoI, 21 (18.1%) had a perpendicular AoI, and 26 (22.4%) had an up/down AoI. Table 1 presents the FR and GLS values for each AoI category. Statistical analysis revealed significant differences in FR between the different AoI categories (p = 0.004 and p = 0.029, respectively).

### Mean GLS values and the influence of AoI

Table 2 presents the descriptive statistics of GLS for the three AoI categories. There was no statistically significant difference between AoI for the mean LV-GLS and RV-GLS values.

### Sensitivity analysis

Tables 3 and S1 present the comparison of GLS measurements obtained from different AoI categories using Semmler's definition for AoI (Fig 1)[13]. The analysis revealed a statistically

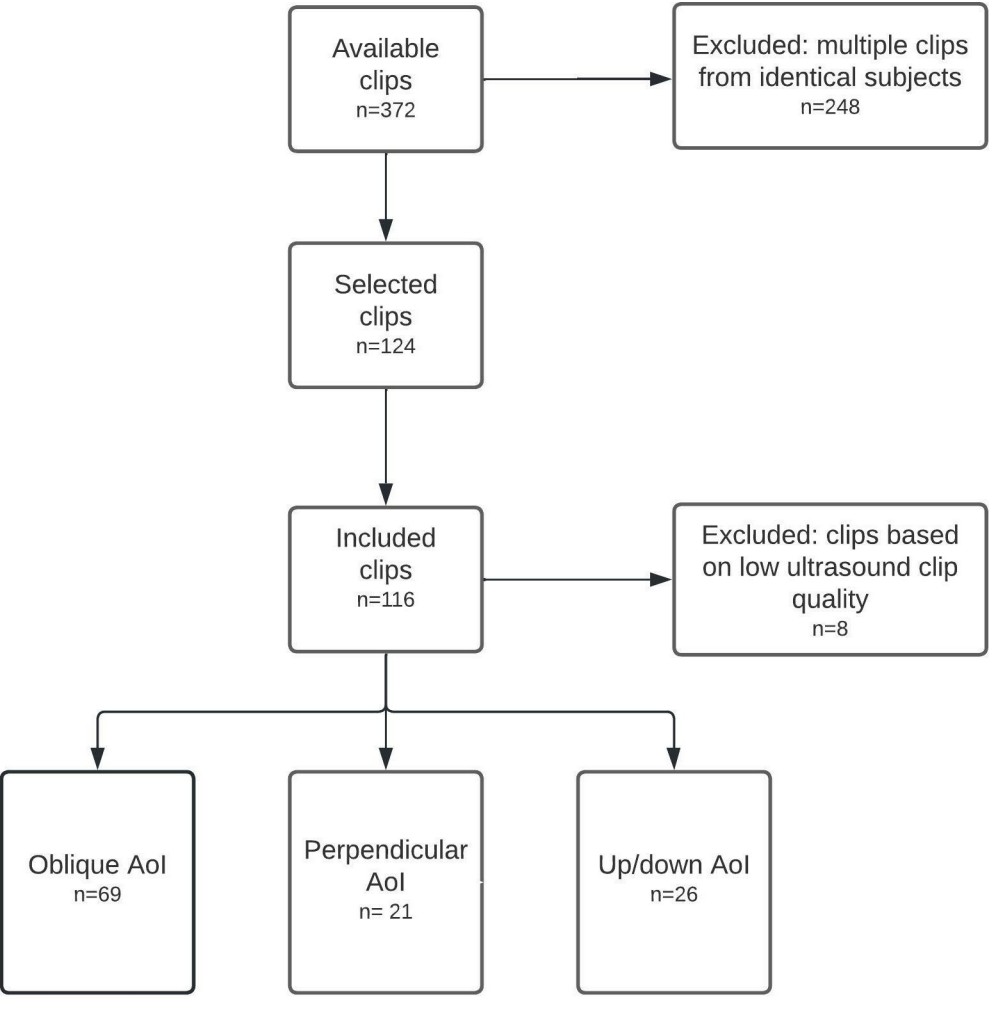

**Fig 2. Flowchart of the fetal heart clips.**

**Table 1. Median frame rates and mean global longitudinal strain per angle of insonation.**

| | Angle of Insonation | | |
| --- | --- | --- | --- |
| | **Oblique** | **Perpendicular** | **Up/down** |
| | *n = 69* | *n = 21* | *n = 26* |
| *Median # frames / second (IQR)* | 119 (109–138) | 115 (90–125) | 97 (89–128) |
| *Median # frames / cardiac cycle (IQR)* | 50 (43–55) | 49 (36–52) | 40 (36–52) |
| *Mean GLS left ventricle (95%CI)* | -22.23 (-24.35 – -20.10) | -24.15 (-28.57 – -19.72) | -19.47 (-21.83 – -17.12) |
| *Mean GLS right ventricle (95%CI)* | -20.57 (-22.28 – -18.87) | -21.87 (-24.83 – -18.91) | -19.30 (-21.44 – -17.16) |

IQR = interquartile ranges, 95%CI = 95% confidence interval.

**Table 2. Mean difference of GLS between different angles of insonation.**

| | Reference Angle of Insonation | Comparison Angle of Insonation | Mean difference (95% CI) | P-value* |
| --- | --- | --- | --- | --- |
| **Left ventricle** | Oblique | Perpendicular | 1.92 (-2.92 – 6.76) | 0.425 |
| | Oblique | Up/down | -2.75 (-5.87 – 0.36) | 0.082 |
| | Perpendicular | Up/down | -4.67 (-9.58 – 0.24) | 0.062 |
| **Right ventricle** | Oblique | Perpendicular | 1.30 (-2.05 – 4.65) | 0.435 |
| | Oblique | Up/down | -1.28 (-3.97 – 1.41) | 0.345 |
| | Perpendicular | Up/down | -2.58 (-6.12 – 0.97) | 0.149 |

95%CI = 95% confidence interval. * P-values <0.05 are considered statistically significant.

**Table 3. Mean difference of GLS between the angles of insonation as categorized by Semmler et al. [13].**

| | Reference Angle of Insonation | Comparison Angle of Insonation | Mean difference (95% CI) | P-value* |
| --- | --- | --- | --- | --- |
| **Left ventricle** | Oblique | Perpendicular | 0.96 (-6.16 – 8.09) | 0.777 |
| | Oblique | Up/down | -3.83 (-7.49 – -0.17) | **0.041*** |
| | Perpendicular | Up/down | -4.79 (-11.82 – 2.23) | 0.165 |
| **Right ventricle** | Oblique | Perpendicular | 0.64 (-4.88 – 6.16) | 0.084 |
| | Oblique | Up/down | -2.56 (-5.76 – 0.65) | 0.115 |
| | Perpendicular | Up/down | -3.20 (-8.84 – 2.45) | 0.243 |

95%CI = 95% confidence interval. * P-values <0.05 are considered statistically significant.

significant decrease in LV-GLS values for the oblique AoI compared to the up/down AoI (p = 0.041).

## Discussion

In this study, we assessed the differences in fetal LV-GLS and RV-GLS between AoI. No significant differences in GLS were shown between the different AoI. When comparing AoI as defined in literature by Semmler et all, the LV-GLS value of the oblique AoI was significantly decreased (i.e. more negative) compared to the up/down AoI. There was no significant difference in RV-GLS for all analyses.

The assumption that 2D-STE would be angle independent was based on in vitro studies [6,7]. However, in a population of infants between 6 and 18 years old, a difference in GLS values was found between AoI [24]. Only one study by Semmler et all investigated the impact of AoI on GLS in fetuses. This study demonstrated a significantly increased (i.e. less negative)

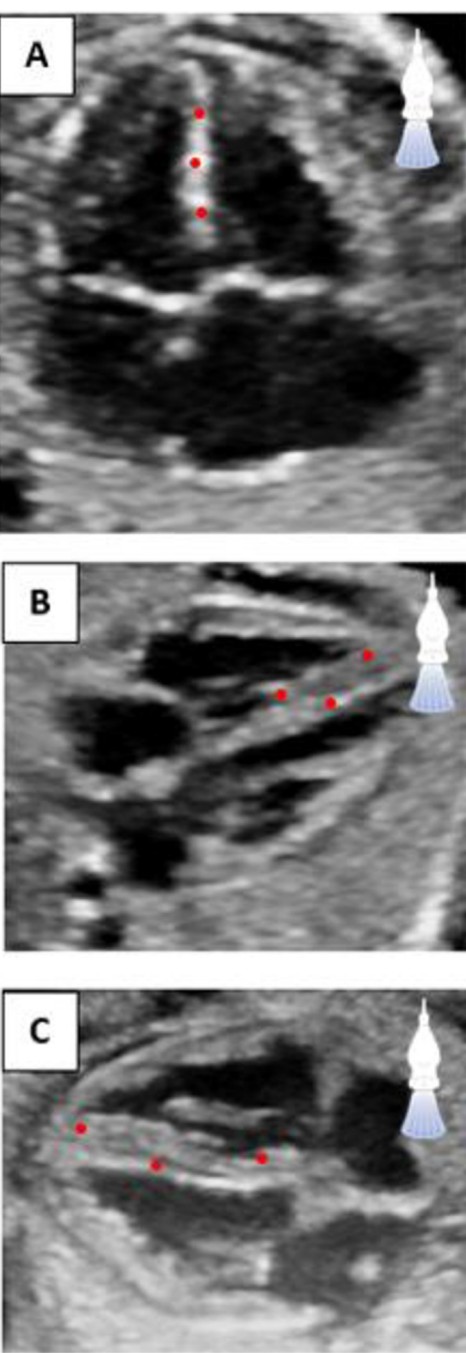

**Fig 3.** (A) Within the up/down angle of insonation, the myocardium is thinly depicted, and the kernels are all in the same axial alignment compared to the ultrasound beams. (B/C) The thickness of the myocardium is fully displayed in the perpendicular and oblique angles of insonation.

LV-GLS value in the up/down AoI compared to the oblique and perpendicular AoI [13]. Our study could only confirm these findings within the sensitivity analysis, showing a significant difference for LV-GLS between the up/down AoI compared to the oblique AoI. However, in our primary analysis, including all angles between the intraventricular septum and the ultrasound beam, no significant difference in GLS between the different AoI was found.

The anisotropic nature of B-mode ultrasound imaging may explain the differences in GLS values between AoI, as found in the literature and our sensitivity analysis. The ability of an ultrasound system to adequately distinguish between two speckles at a given tissue depth may be hampered by low axial and lateral resolution [25]. However, axial and lateral resolution are dictated by the transmit frequency of the transducer, bandwidth and geometry of the ultrasound array. Depending on these features, the center frequency of the probe should be adjusted to optimize resolution and minimize the impact of AoI on strain measurements [14]. However, even with the most optimal settings, a difference between AoI remains due to differences in visualization of the myocardium. Consequently, in the up/down AoI, speckles lie mainly in the same axial alignment as the ultrasound beams, complicating adequate tracing of the speckles even with good resolution (Fig 3) [14]. This may potentially clarify differences in GLS values between AoI.

On the other hand, the transition zones may explain why we did not find a significant difference in GLS between AoI. The transition zones between the intraventricular septum and the ultrasound beam increase differences between AoI and therefore facilitate the detection of differences between AoI [13]. However, in a highly motile fetus, achieving adequate fetal heart clips can already be challenging and time consuming, and excluding certain AoI may lead to a drop in clinical applicability. We suggest using all end-systolic angles between the intraventricular septum and the ultrasound beam for future studies to increase clinical applicability.

Further, higher frame rates are known to result in increased GLS (i.e. more negative GLS values), while low frame rates may cause an underestimation of the impact of AoI on GLS [26,27]. Although fetal heart clips with FR <80 frames per second were excluded in this study, a significantly lower mean FR was found for the up/down AoI compared to the oblique and perpendicular AoI. This might have resulted in an underestimation of GLS in the up/down AoI [27–34]. However, this does not affect the conclusion of this study since the GLS value would decrease (i.e. become more negative) at a higher FR and thus, the difference between AoI would then only decrease.

Results of our study show that differences between AoI can only be observed when the angle between the intraventricular septum and the ultrasound beam are far enough apart. In addition, it remains rather uncertain if differences between AoI have any impact on clinical outcomes. Unfortunately, this study was unable to investigate clinical relevance since the normal values for TomTec software are based on the same ultrasound clips used for the current study [15]. However, the absolute mean GLS values are within a relatively small range, while normal values for the different vendors of ultrasound machines and 2D-STE software have a wider distribution, suggesting that all GLS values are within a normal range irrespective of the AoI [15,16].

Finally, there were no significant differences between AoI in the right ventricle. This might be explained by the anatomical architecture of the right ventricle, including the moderator band and increased trabeculations, that might impede accurate delineation [4]. However, conclusive evidence is lacking, but it was out of scope for this study.

## Strengths and limitations

This was the first study to investigate the impact of AoI in the right ventricle. Furthermore, we only used frame rates ≥80Hz to correct for the high fetal heart rate [8]. We also limited the influence of cardiac maturation by using a narrow GA window [15]. We clearly defined the endocardial border as default ROI [22], we assessed fetal heart clip quality [14] and investigated different definitions for AoI [13].

A limitation is that we have not studied every AoI within a single fetus. Now, differences between AoI may have been masked by inter-fetal differences. Also, fetal GLS values depend

on the used ultrasound machine and 2D-STE software [35,36]. Therefore, the results of this study only apply to TomTec Cardiac Performance 1.2 software and the Philips EPIQ ultrasound system. However, literature also demonstrated an impact of AoI on GLS values using Vitrea Software and the Canon Aplio i800 or Aplio i900 ultrasound machine [13].

## Conclusion

No significant difference in fetal LV- and RV-GLS between the different AoI in fetal 2D-STE was found in this study. However, with other definitions for AoI, a significant difference in LV-GLS was found between de oblique AoI compared to the up/down AoI.

Future research should validate the actual impact of AoI on fetal GLS by generating reference values for all three AoI measured within the same fetus.

## Supporting information

**S1 Table. Median frame rates and mean global longitudinal strain for the sensitivity analysis.**
(DOCX)

## Author Contributions

**Conceptualization:** Thomas J. Nichting, Chantelle M. de Vet, Myrthe van der Ven, Daisy A. A. van der Woude, Marta Regis, Ruud J. G. van Sloun, S. Guid Oei, Judith O. E. H. van Laar, Noortje H. M. van Oostrum.

**Data curation:** Thomas J. Nichting, Chantelle M. de Vet, Noortje H. M. van Oostrum.

**Formal analysis:** Thomas J. Nichting, Chantelle M. de Vet.

**Investigation:** Thomas J. Nichting, Chantelle M. de Vet.

**Methodology:** Thomas J. Nichting, Chantelle M. de Vet, Myrthe van der Ven, Daisy A. A. van der Woude, Marta Regis, Ruud J. G. van Sloun, S. Guid Oei, Judith O. E. H. van Laar, Noortje H. M. van Oostrum.

**Project administration:** Thomas J. Nichting, Chantelle M. de Vet.

**Supervision:** Myrthe van der Ven, Daisy A. A. van der Woude, S. Guid Oei, Judith O. E. H. van Laar, Noortje H. M. van Oostrum.

**Visualization:** Thomas J. Nichting, Chantelle M. de Vet.

**Writing – original draft:** Thomas J. Nichting, Chantelle M. de Vet.

**Writing – review & editing:** Thomas J. Nichting, Chantelle M. de Vet, Myrthe van der Ven, Daisy A. A. van der Woude, Marta Regis, Ruud J. G. van Sloun, S. Guid Oei, Judith O. E. H. van Laar, Noortje H. M. van Oostrum.

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
