## [Decision Letter · Decision Letter 0]

14 Apr 2023

PONE-D-23-04326The impact of angles of insonation on left and right ventricular global longitudinal strain estimation in fetal speckle tracking echocardiography.PLOS ONE

Dear Dr. Nichting,

Thank you for submitting your manuscript to PLOS ONE. After careful consideration, we feel that it has merit but does not fully meet PLOS ONE’s publication criteria as it currently stands. Therefore, we invite you to submit a revised version of the manuscript that addresses the points raised during the review process.

We look forward to receiving your revised manuscript.

Kind regards,

Zahra Hoodbhoy

Academic Editor

PLOS ONE

Additional Editor Comments:

Reviewer 1:

The study is interesting and written comprehensively enough to understand. The objectives, and methodology have provided adequately that the paper does offer enough details to reproduce the experiments.

No major revision from my side. However, there are few comments.

1) In abstract, under objectives, first few lines seem background. So should come under heading of it.

2) The objective need to be revised (as primary and secondary) as author studied the Semmler et al. definition separately by sensitivity analysis.

3) Similarly in abstract, under method, the last lines need to be revised as creating confusion to reader, please define clearly, why and to whom you did sensitivity analysis?

4) In manuscript, under method it was mentioned that ‘Women who developed gestational diabetes or hypertensive pregnancy disorders, and those who gave birth to a neonate with a birth weight below the 10th percentile or with congenital or genetic abnormalities were excluded from the analysis as these factors may influence myocardial deformation values’.

I did not find any point where it is mentioned that clips were selected after the birth of baby. If this is so, please mention it otherwise remove this statement.

5) Under data acquisition, these lines seem accessary and should be removed ‘Adequate quality fetal heart clips can be achieved by an appropriate setting of the region of interest (ROI)8, and optimal settings of depth, width, and zoom box help to achieve maximum FR, while not critically degrading image quality minimizing the impact of potential speckle anisotropy on fetal 2D-STE.

Add objective points on which you checked image quality, via in a table or here in text.

6) Under angle of Insonation 2nd paragraph needs to be rephrased.

7) Results are not written in a very structured manner, please revise it.

Reviewer 2:

The article is very well conceptualized. the authors have explained the process of speckle tracking frame rate and GLS which are actually not easy to understand in simplified way. They have already discussed their limitation in terms of retrospective data and unavailability of normal values, but articles with scientific content like these should be encouraged as these will set the trend for more work and normal standards. i must congratulate the authors on their extremely important and hard work.

Reviewers' comments:

Reviewer's Responses to Questions

**Comments to the Author**

1. Is the manuscript technically sound, and do the data support the conclusions?

Reviewer #1: Yes

Reviewer #2: Yes

2. Has the statistical analysis been performed appropriately and rigorously? 

Reviewer #1: Yes

Reviewer #2: Yes

3. Have the authors made all data underlying the findings in their manuscript fully available?

Reviewer #1: Yes

Reviewer #2: Yes

4. Is the manuscript presented in an intelligible fashion and written in standard English?

Reviewer #1: Yes

Reviewer #2: Yes

5. Review Comments to the Author

Reviewer #1: Manuscript ID: PONE-D-23-04326

The impact of angles of insonation on left and right ventricular global longitudinal strain estimation in fetal speckle tracking echocardiography.

The study is interesting and written comprehensively enough to understand. The objectives, and methodology have provided adequately that the paper does offer enough details to reproduce the experiments.

No major revision from my side. However, there are few comments.

1) In abstract, under objectives, first few lines seem background. So should come under heading of it.

2) The objective need to be revised (as primary and secondary) as author studied the Semmler et al. definition separately by sensitivity analysis.

3) Similarly in abstract, under method, the last lines need to be revised as creating confusion to reader, please define clearly, why and to whom you did sensitivity analysis?

4) In manuscript, under method it was mentioned that ‘Women who developed gestational diabetes or hypertensive pregnancy disorders, and those who gave birth to a neonate with a birth weight below the 10th percentile or with congenital or genetic abnormalities were excluded from the analysis as these factors may influence myocardial deformation values’.

I did not find any point where it is mentioned that clips were selected after the birth of baby. If this is so, please mention it otherwise remove this statement.

5) Under data acquisition, these lines seem accessary and should be removed ‘Adequate quality fetal heart clips can be achieved by an appropriate setting of the region of interest (ROI)8, and optimal settings of depth, width, and zoom box help to achieve maximum FR, while not critically degrading image quality minimizing the impact of potential speckle anisotropy on fetal 2D-STE.

Add objective points on which you checked image quality, via in a table or here in text.

6) Under angle of Insonation 2nd paragraph needs to be rephrased.

7) Results are not written in a very structured manner, please revise it.

My Best

Reviewer #2: The article is very well conceptualized. the authors have explained the process of speckle tracking frame rate and GLS which are actually not easy to understand in simplified way. They have already discussed their limitation in terms of retrospective data and unavailability of normal values, but articles with scientific content like these should be encouraged as these will set the trend for more work and normal standards. i must congratulate the authors on their extremely important and hard work.

6. PLOS authors have the option to publish the peer review history of their article (what does this mean?). If published, this will include your full peer review and any attached files.

Reviewer #1: **Yes: **Fatima Ali

Reviewer #2: **Yes: **Shazia Mohsin

---

## [Author Response · Author response to Decision Letter 0]

18 May 2023

Máxima MC, Veldhoven

Department of Gynaecology and Obstetrics

P.O. box 7777, 5500MB Veldhoven, The Netherlands

14 May 2023

Veldhoven, The Netherlands

Subject: Rebuttal letter revisions PONE-D-23-04326

Dear academic editor Zahra Hoodbhoy,

We hereby seek to submit the revised version of the manuscript, addressing the points brought forth during the review process for the article entitled "The impact of angles of insonation on left and right ventricular global longitudinal strain estimation in fetal speckle tracking echocardiography". Enclosed below, you will find a comprehensive response addressing each individual point raised by the reviewers.

Comments of the reviewers

Reviewer 1

The impact of angles of insonation on left and right ventricular global longitudinal strain estimation in fetal speckle tracking echocardiography. The study is interesting and written comprehensively enough to understand. The objectives, and methodology have provided adequately that the paper does offer enough details to reproduce the experiments. No major revision from my side. However, there are few comments.

Answer: Thank you for your positive feedback on the study. We are glad to note that you found the article interesting and that it was written in a comprehensive manner, with sufficient details for experiment reproduction. It's also encouraging to know that no significant revisions are required on your part. We appreciate your comments and are happy to discuss the comments below to further improve the article.

1. In abstract, under objectives, first few lines seem background. So should come under heading of it.

Answer: Thank you for your comment. Regarding the abstract, we agree that the first few lines of the objectives section may be more appropriate as background information. However, the submission guidelines of PLOS One specifically require the heading "Objectives" in the abstract. Considering these guidelines, we have decided to maintain the current structure of the manuscript and keep the objectives section as it is.

2. The objective need to be revised (as primary and secondary) as author studied the Semmler et al. definition separately by sensitivity analysis.

Answer: We appreciate your valuable comment. We have recognized the need for greater accuracy in addressing the objective of the study as we study the definition for AoI by Semmler et al. separately by doing a sensitivity analysis. As you suggested, we have divided the objective into a primary and secondary objective. The changes reflecting this comment can be found in the revised manuscript on page 3, line 55. 

3. Similarly in abstract, under method, the last lines need to be revised as creating confusion to reader, please define clearly, why and to whom you did sensitivity analysis?

Answer: As we revised the objective, we have removed this sentence from the manuscript as it did not longer provide any additional information. Further elaboration on the rationale behind the various definitions can be found in the discussion section on page 15, starting at line 284.

4. In manuscript, under method it was mentioned that ‘Women who developed gestational diabetes or hypertensive pregnancy disorders, and those who gave birth to a neonate with a birth weight below the 10th percentile or with congenital or genetic abnormalities were excluded from the analysis as these factors may influence myocardial deformation values’.

I did not find any point where it is mentioned that clips were selected after the birth of baby. If this is so, please mention it otherwise remove this statement.

Answer: Thank you for bringing this to our attention. The confirmation of the exclusion criteria took place 10 weeks after childbirth. We have made the necessary update to the manuscript to clarify the timing of the confirmation of the exclusion criteria. On page 7, line 123 we have included an additional sentence to address this aspect.

5. Under data acquisition, these lines seem accessary and should be removed ‘Adequate quality fetal heart clips can be achieved by an appropriate setting of the region of interest (ROI)8, and optimal settings of depth, width, and zoom box help to achieve maximum FR, while not critically degrading image quality minimizing the impact of potential speckle anisotropy on fetal 2D-STE. Add objective points on which you checked image quality, via in a table or here in text.

Answer: Thank you for your valuable feedback. In the literature, image quality for adequate fetal heart clips is assessed using these descriptive an subjective measures, while frame rates serve as the sole objective measure. In an effort to enhance clarity, we have rewritten the paragraph to provide a more concise explanation. Changes can be found in the revised manuscript on page 7, line 140.

6. Under angle of Insonation 2nd paragraph needs to be rephrased.

Answer: In the second paragraph of the angle of insonation section, we have rephrased the content to improve its clarity and coherence.

7. Results are not written in a very structured manner, please revise it.

Answer: Thank you for your feedback. To address the concern regarding the structure of the results section, we have revised it to enhance its organization and coherence.

Reviewer 2

The article is very well conceptualized. the authors have explained the process of speckle tracking frame rate and GLS which are actually not easy to understand in simplified way. They have already discussed their limitation in terms of retrospective data and unavailability of normal values, but articles with scientific content like these should be encouraged as these will set the trend for more work and normal standards. I must congratulate the authors on their extremely important and hard work.

Answer: Thank you for your feedback on the article. We appreciate your positive assessment of its conceptualization and the authors' ability to simplify complex concepts like speckle tracking frame rate and GLS for a wider audience. With the acknowledgment of study limitations we aim to achieve transparency and a responsible approach, providing readers with a comprehensive understanding of the research's scope and implications. We agree with your sentiment that encouraging articles like these is important. They contribute to scientific advancement and establish standards for future work.

We hope that this letter gives a sufficient answer to all the comments. If we can provide you with any further information, do not hesitate to reach us.

Yours sincerely, 

On behalf of all authors,

Drs. T.J. Nichting

Corresponding author

Mail: Thomas.nichting@mmc.nl

Phone: +31 6 48649152

---

## [Decision Letter · Decision Letter 1]

29 May 2023

The impact of angles of insonation on left and right ventricular global longitudinal strain estimation in fetal speckle tracking echocardiography.

PONE-D-23-04326R1

Dear Dr. Thomas Johannes Nichting

We’re pleased to inform you that your manuscript has been judged scientifically suitable for publication and will be formally accepted for publication once it meets all outstanding technical requirements.

Kind regards,

Zahra Hoodbhoy

Academic Editor

PLOS ONE

Additional Editor Comments (optional):

Reviewers' comments:

Reviewer's Responses to Questions

**Comments to the Author**

1. If the authors have adequately addressed your comments raised in a previous round of review and you feel that this manuscript is now acceptable for publication, you may indicate that here to bypass the “Comments to the Author” section, enter your conflict of interest statement in the “Confidential to Editor” section, and submit your "Accept" recommendation.

Reviewer #1: All comments have been addressed

2. Is the manuscript technically sound, and do the data support the conclusions?

Reviewer #1: Yes

3. Has the statistical analysis been performed appropriately and rigorously? 

Reviewer #1: Yes

4. Have the authors made all data underlying the findings in their manuscript fully available?

Reviewer #1: Yes

5. Is the manuscript presented in an intelligible fashion and written in standard English?

Reviewer #1: Yes

6. Review Comments to the Author

Reviewer #1: The study is very interesting and would be beneficial for readers to replicate. The objectives and methods were well defined. The study should not submit in other journal, which is against publication ethics.

7. PLOS authors have the option to publish the peer review history of their article (what does this mean?). If published, this will include your full peer review and any attached files.

Reviewer #1: **Yes: **Fatima Ali

---

## [Editor Report · Acceptance letter]

4 Jul 2023

PONE-D-23-04326R1 

The impact of angles of insonation on left and right ventricular global longitudinal strain estimation in fetal speckle tracking echocardiography. 

Dear Dr. Nichting:

I'm pleased to inform you that your manuscript has been deemed suitable for publication in PLOS ONE. Congratulations! Your manuscript is now with our production department. 

Kind regards, 

on behalf of

Dr. Zahra Hoodbhoy 

Academic Editor

PLOS ONE